# Measurements of Dynamic Deformations of Building Structures by Applying Wire Sensors

**DOI:** 10.3390/s19020255

**Published:** 2019-01-10

**Authors:** Grzegorz Cieplok, Łukasz Bednarski

**Affiliations:** Faculty of Mechanical Engineering and Robotics, AGH University of Science and Technology, 30 Mickiewicza Avenue, 30-059 Krakow, Poland; lukaszb@agh.edu.pl

**Keywords:** vibrating wire sensors, measurements of deformations, building structures, dynamic measurements

## Abstract

Measurements of deformations by means of vibrating wire sensors are very important in the monitoring of building structures. These types of sensors are characterized by a high resistance to environmental conditions, long time of measurement stability, and a possibility to use long electric cables with a solid encasement in concrete. Vibrating wire sensors are mainly used for measuring stable or slowly changing deformations, however applications of these sensors for measuring time-variable deformations are becoming popular. New solutions generate new problems, which in case of vibrating wire sensors are mainly related to the operational stability of the systems exciting wire vibrations. The structure of such sensors and the length of the electric cables, which can reach a few kilometers, have an essential influence on their operations. This paper undertakes the task of determining the influence of the electric cables length on the proper operation of the measurement system and provides advice for improvements of its measurement possibilities. The subject of investigation constitutes a measurement system based on a self-exciting impulse exciter, for which the impedance of the electric cables and of the vibrating wire sensor are the most essential parameters. A mathematical model of this system, experimental verification of the model, and the results of theoretical analyses and measurement tests for electric cables of various lengths are presented in this paper.

## 1. Introduction

Vibrating wire sensors, on account of their principle of operation, are mainly intended for measuring constant or slowly changing deformations [1,2]. They consist of a base, two catches, a vibrating wire, an electromagnet, and electronic systems (Figure 1). In typical arrangements the electromagnet functions as the exciter of vibrations as well as the sensor reading its motion [3,4,5,6].

The wire is excited for movement by means of magnetic field impulses generated by the electromagnet and, after sounding out the transient states, performs natural vibrations. On the basis of the vibrations, it is possible to determine the frequency and tension of the wire and then its relative elongation. On account of the damped character of vibrations, the excitation process of the wire is done in time interval cycles, as long as vibrations of successive cycles are not overlapping. Due to this, deformations can only be determined pointwise over intervals of a few to a dozen or so seconds.

Vibrating wire sensors have applications in the technical state control of bridges [7,8], ground anchor systems [9], tunnels, large scale buildings [10,11,12], and transport infrastructure [13]. The technique of vibrating wire measurements has become an important branch of the metrology applied in measurements of elements’ displacements relative to each other, deformations, stresses [14], changes of fugue cleavages, changes in the tilt of building objects, the subsidence of pillars, the ground pressure on structure elements, etc.

The needs of designers, contractors, and object operators show that static measurements are not enough. When loads are dynamic (vehicles, wind, earthquakes) it becomes necessary to conduct measurements where sensors allow one to obtain a response, of the variability of the measured values, at least 100 times per second. The measured strain histogram, made on the basis of high frequency measurements, can be the basis for fatigue analysis. Such analysis is necessary to assess the durability of heavily used structures exposed to dynamic loads, e.g., bridges [8].

In the case of systems with continuous measurements of deformations, the situation is more complex. These types of systems have to set the wire in non-fading away vibrations without disturbing its natural frequency.

The vibrating wire sensor and the transducer system of the Norvegian Geotechnical Institute can be assumed as the first such solution. The principle of operation of this system is presented in Reference [15]. The system consists of two coils, the drive coil in the transistor collector circuit and the pickup coil connected to the joint base emitter, functioning as feedback. The system of joints allows one to excite self-exciting vibrations at the frequency corresponding to the first form of the natural frequencies of the wire. The application of the transistor directly at the transducer coils allowed, according to the authors of the approach, for stable transducer operation and transmission of the measuring signal up to distances of 12 km.

Devices CDM-VW300 and CDM-VM305 of the Campbell Scientific Company can be used for non-stationary vibrations where the frequency of the measurement samples is converted from 20 to 333 Hz. These devices are based on patent descriptions No. US 7779690 and US 8671758 [16,17]. In order to accurately determine the natural frequencies of the wire, the discrete Fourier transform and spectral approximation were used. This solution is also characterized by very good filtration possibilities, which allows one to use long, up to a dozen or so kilometers, electric cables connecting the sensor with the control system. On account of the Fourier transform applied in these inventions, which requires measuring samples for a properly long durability time, these solutions (using one electromagnet) should be classified as solutions with alternate excitation and wire vibrations recording.

Both solutions have certain disadvantages. In case of the exciter of the Norvegian Geotechnical Institute, the presence of semiconductor elements excludes applications of sensors in concrete structures where, during building and in further exploitations, electronic elements are highly subjected to damage. On the other hand, based on the experiences of users of the Campbell Company devices, devices subjected to step loads can lose tuning of the wire vibrations and fall out of synchronization. The significant cost of the device is not without meaning.

Another class of solutions for vibrating measurement are high precision impedance-frequency transducers (capacitive or inductive) which use quartz crystals to compensate temperature drift, have high stability and fast response, and can be modulated with very low frequency. These methods are described in References [18,19].

Two new solutions of vibrating wire sensors, which undertake the task of meeting the requirements of building engineering are described in References [20,21,22]. Both solutions utilize self-exciting vibrations, the first one is based on the van der Pol generator, while the second one on the impulse exciter. Both solutions passed positively laboratory tests. Due to the application of common and generally available vibrating wire sensors for the impulse exciter, this solution has become the more practical one and holds promise for industrial implementations.

Natural vibrations of the wire that don’t fade away are generated by the electromagnet, whose supply is cyclically interrupted in a way that assures the creation of self-excited vibrations of the wire (Figure 2). The essence of the operation consists of determining the way to supply the main electromagnet coil $L_{2}$, which is excited only during the semi-periods in which the wire is moving closer to the electromagnet. In the remaining semi-periods, i.e., when the wire is receding from the electromagnet, the electromagnet is disconnected from the supply. The direction of the wire movement, needed to properly determine the semi-period of the electromagnet operation, is determined by the voltage induced in the pickup coil $L_{1}$. The interrupted operations of the electromagnet ensure that the wire is entering into a self-excited vibration state, characterized by a limit cycle with constant amplitude of vibrations velocity and with a frequency equal to the natural vibration frequency.

The voltage $U_{p}$ induced in the pickup coil, compatible in phase with the wire vibrations velocity, is transferred to the input of the preliminary transformation system U1, whose task is to increase the signal to the operation level of the electronic systems and filter out disturbances. The preliminarily transferred signal $U_{f}$ is directed to the electronic comparator U2, which is aimed at capturing the negative value of the voltage $U_{f}$ corresponding to the wire movement in the electromagnet’s direction. The output voltage of the comparator $U_{0}$ determined by two states, high for negative values of $U_{f}$ and low for positive values, is directed to the input of the electronic switching system U3, based on a simple circuit consisting of one transistor (Figure 9 in Reference [22]).

The task of the switching system is to close and open—in agreement with the voltage—the electromagnet $L_{2}$ current circuit in order to excite the wire movement. The operation cycle ensures the wire is driven in the electromagnet direction to increase its kinetic energy and a subsequent natural and free reverse movement. Due to small damping, there is no danger of wire vibrations fading away in the semi-periods during which the electromagnet is switched off.

The exciter, when subjected to step and impulse loads, operated properly; it did not lose synchronicity and was able to determine the instantaneous state of the wire elongation and the sensor base deformation from the wire vibrations frequency. The situation changes when the length of the electric cables is increased to a few hundred meters. At that length, the exciter loses rhythm, is no longer in agreement with the wire motion, and thus loses synchronicity.

## 2. Formulating the Model of a Vibrating Wire Sensor with Electric Cables

On the basis of preliminary measurements of the electric cable parameters, it was found that the parameters most subjected to changes of length are the longitudinal resistance and the transverse capacitance of the cables. On account of this, a two-port resistance–capacitance network of the T-type was assumed as the model of the electric cables. The scheme of the system, in which the sensor coils, the cables, the source of the drive coil supply, and other equivalent parts of the wire are exposed, is presented in Figure 3. The meaning of the equivalent mass and the remaining equivalent parameters of the vibrating wire are described in detail in Reference [21].

The dynamic equations of motion for the system, given by Equations (Equation 1)–(Equation 5), were determined on the basis of the above-mentioned scheme. The following generalized coordinates were assumed: current $i_{11}$ of coil $L_{1}$, current $i_{21}$ of coil $L_{2}$, voltage $u_{C1}$ on condenser $C_{1}$, voltage $u_{C2}$ on condenser $C_{2}$, and the position coordinate *x* of the vibrating wire equivalent mass:(1)$$\begin{array}{c} L_{1}\left(x\right)\frac{di_{11}}{dt}+i_{11}\frac{dx}{dt}\frac{\partial L_{1}\left(x\right)}{\partial x}+M_{12}\frac{di_{21}}{dt}-a\frac{dx}{dt}+\left(R_{p11}+R_{1}\right)i_{11}+u_{C1}=0 \end{array}$$
(2)$$\begin{array}{c} L_{2}\left(x\right)\frac{di_{21}}{dt}+i_{21}\frac{dx}{dt}\frac{\partial L_{2}\left(x\right)}{\partial x}+M_{12}\frac{di_{11}}{dt}-a\frac{dx}{dt}+\left(R_{p21}+R_{2}\right)i_{21}+u_{C2}=0 \end{array}$$
(3)$$\begin{array}{c} u_{C1}-\left(R_{d}+R_{p12}\right)\left(i_{11}-C_{1}\frac{du_{C1}}{dt}\right)=E \end{array}$$
(4)$$\begin{array}{c} u_{C2}-\left(R_{0}+R_{p22}\right)\left(i_{21}-C_{2}\frac{du_{C2}}{dt}\right)=0 \end{array}$$
(5)$$\begin{array}{c} m\frac{d^{2}x}{dt^{2}}+b\frac{dx}{dt}+kx=\frac{1}{2}\frac{\partial L_{1}\left(x\right)}{\partial x}i_{11}^{2}+\frac{1}{2}\frac{\partial L_{2}\left(x\right)}{\partial x}i_{21}^{2} \end{array}$$

The electrical parameters of the system were determined by direct measurements and are listed in Table 1. The only exception constitutes the mutual inductance $M_{12}$, which was experimentally selected by matching the real waveforms recorded by the oscilloscope with theoretical waveforms obtained by numerical simulations of the equations of motion.

The waveforms, mentioned above, are presented in Figure 4 and Figure 5. As can be seen, the results are in good agreement with each other and thus, the formulated model of the electric part is the basis for further analyses.

The parameters of the mechanical part were determined on the basis of the natural frequencies of the vibrating wire. In order to do this, the sensor housing was cut and the wire was jerked. This jerking causes wire movement, which—due to the damping—took the form of natural damped vibrations, Figure 6. On the basis of the logarithmic damping decrement of vibrations $\delta =\ln(\frac{A_{1}}{A_{n}})=1.97$, the damping coefficient was determined:(6)$$b=\frac{2m\delta }{nT}=9.5\times 10^{-5}\;\;\left(Ns/m\right)\;.$$
On the basis of the frequency of natural vibrations $f_{0}=874.6$ (Hz), the coefficient of elasticity was determined:(7)$$k={\left(2\pi f_{0}\right)}^{2}m=558.3\;\;\left(N/m\right)\;.$$

The equivalent mass of the wire $m=1.85\times 10^{-5}$ (kg) was determined on the basis of its geometrical parameters. It can be noticed, that the relative coefficient of damping of the wire motion ξ=0.00046 is more than twenty times smaller than the typical coefficients for suspensions made of steel elements.

The constant *a* of the sensor, connecting the voltage induced in the pickup coil with the wire movement velocity, was assessed on the basis of Equation (Equation 8):(8)$$v_{0}=c\frac{i_{110}^{2}}{\pi b},$$
which was derived in Reference [22], and where $i_{110}$ marks the steady-state of the current $i_{11}$ in the semi-period after switching on the power supply. This equation allows one to determine the amplitude of the vibration velocity of the wire supplied by a set value of voltage when combined with the amplitude of voltage measured on the coil winding. A value of a=0.0526(Vs/m) was obtained in this way.

## 3. Influence of Disturbances on the Synchronization of the Impulse Exciter

The proper operation of the system under the field conditions depends on several factors. These include: the lengths of electric cables, interactions between the drive coil and pickup coil circuits, and any electrical disturbances. In order to determine the influence of the cable lengths on exciter operation, Equations (Equation 1)–(Equation 5) were applied. Substituting in the autonomic source $E=E_{0}\mathrm{sign}\left(\omega t\right)$ for the controlled source $E=E_{0}\cdot \mathrm{sign}\left(U_{R0}\right)$, a model of an ideal non-inertial exciter was obtained. The response of the control signal is achieved without a time delay in this exciter.

To be able to clearly differentiate the influences of individual factors on the exciter operations, a series of simulation investigations was performed, starting from the most idealized case and moving to more complicated cases, in which strong complex intercoil couplings were analyzed.

### 3.1. Idealized System

The idealized system is considered as the simplest system for analysis, which realizes self-exciting vibrations—without disturbances—as described in the Introduction. It consists of a voltage source controlled by the pickup coil voltage, electric cables, and coils of wire sensors in which the mutual coupling was eliminated.

The results of five simulations for five different length of cables: 500 m, 1 km, 2 km, 4 km, and 6 km are presented in Figure 7. As can be noticed, for the first four distances the system was self-exciting and obtained vibrations at a frequency of 874.8 Hz, corresponding precisely to the wire’s natural vibrations. For the first two distances, compatibility between the wire vibration amplitude and the theoretical Equation (Equation 8) was also obtained.

Table 2 presents an adequate comparison. Of course, the mentioned equation does not take into consideration eventual phase shifts of the current in relation to the vibrating wire movement, current distortions caused by the presence of inductive and capacitive elements, and voltages induced by the vibrating wire motion. The decrease in the amplitude of the wire vibrations was caused by the decrease in the current exciting the drive coil, on account of the increase in the resistivity of the electric cables.

The phase shift of the current and wire waveforms also has meaning, it is larger for longer cables. The waveforms of current $i_{11}$ and the velocity *v* of the wire middle point for three characteristic lengths, are presented in Figure 8. For cables of length 1 km (Figure 8a), the shift was negligible, while in case of cables of the length of 4 km (Figure 8b) the shift was clearly visible. The moments when the current switched on were displaced in time relative to the moments when the velocity course changed from positive to negative values. This also affected (and to a higher degree) the moments when the current switched off. Such a situation is the reason why the time over which the wire accelerated in the proper (negative) direction was shortened, while the time over which the wire should have been performing free motion (in positive direction) was consumed for its braking.

The situation deepened with increased cable length to such degree that after exceeding a certain length, total decay of the wire free movement occurred and the coil current entered into a pulsating direct character. Such a case is shown in Figure 8c. The frequency of wire vibrations stabilized at a value of 831 Hz and did not correspond to the natural vibration frequency. This motion was the result of excitations of the electro-mechanical system vibrations and could not be the basis for determining the wire longitudinal strain. On account of the impossibility of determining the value $i_{110}$ for l=6 km, the theoretical values of the wire velocity are not included in Table 2.

### 3.2. Influence of Intercoil Coupling

The factor with the worst influence on system operation is the inductive coupling of the drive and pickup coils. On account of current changes, caused by step changes of the supply voltage, a high amplitude electric impulse is induced in the pickup coil. This impulse can exceed several times the value of the voltage induced by the wire movement. In such a situation, vibrations can be excited by the signal depending on the parameters of the electrical part. This case is presented in Figure 9a, where the wire vibrations stabilized at the frequency of approximately 8.25 kHz and at a very low level. In order to recognize the essence of the system’s excitation by the over-voltage signal, a simulation of the same case but with a wire initial velocity of v(0)=2.0 (m/s), is shown in Figure 9b. Voltage oscillations of capacitor $C_{2}$ immediately after the voltage passes the zero value, are seen clearly in this figure. These oscillations are, in turn, the cause of unwanted switching of the supply source. The result of these switchings can be observed on the voltage waveforms for the capacitor $C_{1}$.

The unwanted influence of the intercoil coupling can be decreased by damping the impulses of the pickup coil. The simplest way of this realization is by introducing a bypass capacitor shunting the reference resistor $R_{0}$ of the value, ensuring the short-circuit has fast-changing waveforms. The electric scheme of such solution is shown in Figure 10. On the basis of several tests, a capacitance of $C_{3}=470$ nF, which most efficiently damped over-voltages and does not cause simultaneous essential shifts between the current induced by the wire movement and its velocity, was selected.

The modification of the system essentially improved its operational stability and resistance to over-voltage signals. Simulations performed for a coupling of 30% of the real value (Table 1) were very good. The system was automatically starting up and obtaining vibrations of frequencies of the wire’s natural frequency within a wide range of cables length.

The simulation results for 1 km long cables are presented in Figure 11a,b. The presence of the current impulse induced in the pickup coil, which strongly deforms the sinusoidal waveform of the current induced by the wire movement and the waveform smoothed by the filtrating capacitor $C_{3}$, are seen in these figures.

The application of the bypass capacitor also allowed us to perform the simulation for the total (real) coupling. However, in this case it was necessary to set up the initial velocity of the wire, since without it the system was self-exciting due to the electrical frequency. The simulation results of the sensor subjected to loads, for 1 km long cables, are shown in Figure 12. The sensor load was simulated by a step change of the elasticity coefficient, according to the sequence presented in Table 3. Values of the wire’s natural frequencies, determined on the basis of the theoretical equation $f_{0}=\frac{1}{2\pi }\sqrt{\frac{k}{m}}$, are also given in this table.

It can be noticed that when the system started operations, the wire velocity was systematically increased to the operation point value, and load step changes did not cause the wire to fall out of its natural vibration frequency, corresponding to the given value of the elasticity coefficient.

## 4. Experimental Tests

In order to verify the system properties resulting from theoretical analyses, experimental tests were performed. These tests applied the impulse exciter described in Reference [22], a vibrating wire sensor from the Geokon Company, and electric cables BIT-sensor PE–PVC, as shown in Figure 13. The exciter was subjected to small modifications in order to adjust its operations to experimental conditions. The resistance value $R_{d}$ was changed from 560Ω to 160Ω and the low-pass filter was tuned. The system behavior was tested for various lengths of electric cables, including: 1.5 m, 20 m, 40 m, 100 m, 160 m, and 500 m.

The oscilloscope screen during the wire vibrations recorded for 1.5 m long cables, are presented in Figure 14. Waveforms of the input voltage (from the pickup coil), the input voltage after filtration by means of the internal low-pass filter, and the voltages on the comparator and the exciter output (on the cable end of the drive coil), are shown there. All of the features observed in theoretical waveforms are also found in the oscillographs, including: the high value of the over-voltage impulse, the low value of the voltage induced by the wire movement, and the voltage impulse on the drive cable. In Figure 15, waveforms for 160 m long cables are presented. It is possible to notice the characteristic oscillations of intercoil over-voltage at the electrical system frequency, which grow as the cable length increases, leading to the system’s excitation at the electrical frequencies. Waveforms after the application of the bypass capacitor $C_{3}$, are seen in Figure 15b. Analogous with the simulation investigations, this application decreased the over-voltage values, allowing for more stable system operations. In the case of 500 m long cables, the situation was more spectacular. Without the capacitor the system was not able to keep the wire’s natural frequency and gradually passed to vibrations of electrical frequencies. After the capacitor introduction, the response was stable at the natural frequency for the widest range of wire length changes of all performed tests, Figure 16.

## 5. Conclusions

The analysis of cooperation between the vibrating wire sensor and the impulse exciter performed in this paper indicates that the dynamic states of the circuit sensor and the electric cables are the most important effects for proper operation of the system. The possibility of excitations of vibrations at the wire’s natural frequency depends on the impedance parameters of the sensor coils and electrical parameters of the electric cables. The maximal length of these cables, for which self-excited vibrations will be kept, depends on these parameters. Theoretical analysis of the ideal case indicates that when typical components available in the market are used, operation of the system with electric cables up to 3–4 km long, is possible. However, the ideal case does not taking into account the intercoil coupling, the most essential factor influencing the proper operations of the system. This coupling can, on one hand, totally damp the natural vibrations of the wire and, on the other hand, constitute the reason of exciting unwanted electrical vibrations at frequencies of a few kilohertz. In such a case, filtration systems are important, in order to eliminate the influence of the electrical coupling frequency. Already a simple passive resistor–capacitor (RC) filter applied at the exciter input allowed for proper operation of the system.

Laboratory tests carried out for cables of various length confirmed the main conclusions obtained from the theoretical analyses. The experiments also confirmed the possibility of a practical application of the exciter with electric cables of a length up to 500 m (tests were not performed for longer cables). It seems also, that based on the efficiency of the filtering systems and the uncoupling degree of the sensor coils, it is possible—in practice—to get close to the maximal lengths obtained on the basis of the ideal case analysis.

## Figures and Tables

**Figure 1 sensors-19-00255-f001:**
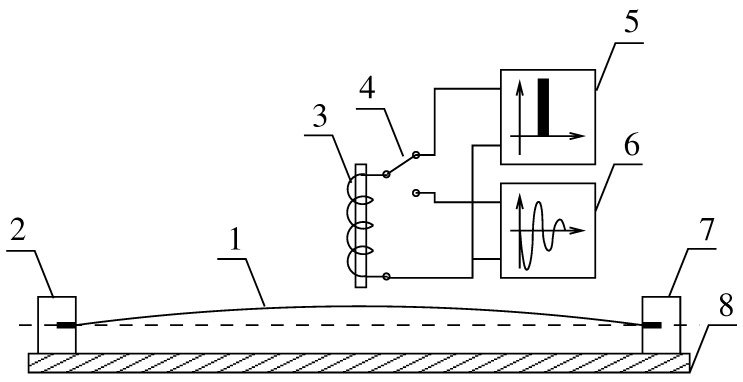
Principle of operation of the strain transducer: 1–wire, 2,7–catches, 3–electromagnet, 4–switch, 5–pulser, 6–analyzer, and 8–base.

**Figure 2 sensors-19-00255-f002:**
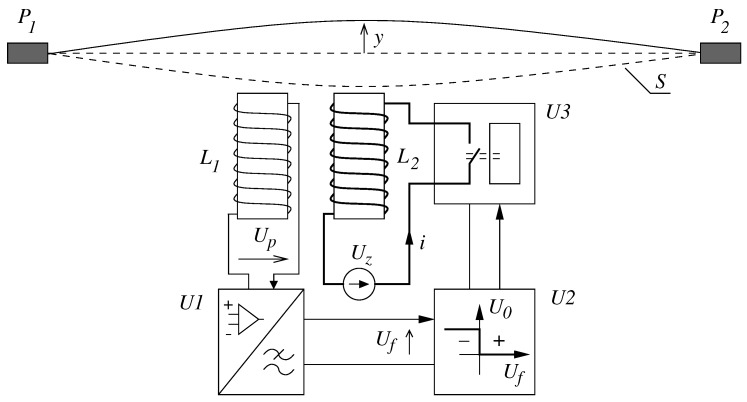
A schematic of the impulse exciter: $L_{1}$–pickup coil, $L_{2}$–drive coil, $U_{z}$–supply, $U_{3}$–interrupter synchronized by the coil voltage signal $L_{1}$, $U_{1}$–low-pass filter, and $U_{2}$–voltage comparator.

**Figure 3 sensors-19-00255-f003:**
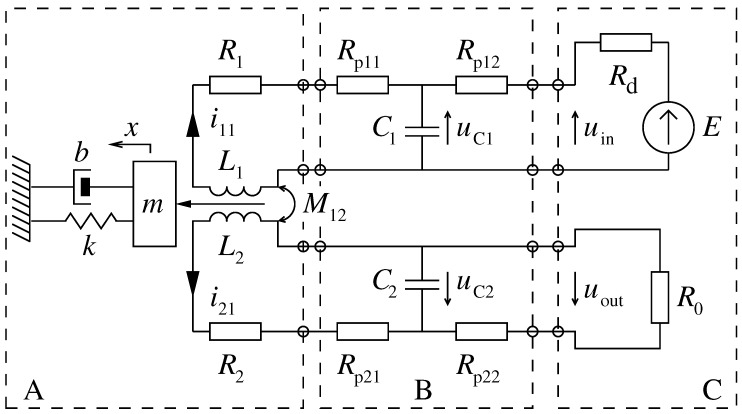
Scheme of the vibrating wire sensor with electric cables. (**A**) Wire sensor section: m,b,k–equivalent wire parameters, $L_{1}$–drive coil inductance, $R_{1}$–drive coil resistance, $L_{2}$–pick up coil inductance, $R_{2}$–pick up coil resistance and $M_{12}$–mutual inductance; (**B**) the cable section: $R_{p11},R_{p12},C_{1}$–a T-type equivalent model of the power cable, and $R_{p21},R_{p22},C_{2}$–a T-type equivalent model of the feedback cable; and (**C**) the power section: *E*–voltage source, $R_{d}$–internal resistance of the source, and $R_{0}$–reference resistance.

**Figure 4 sensors-19-00255-f004:**
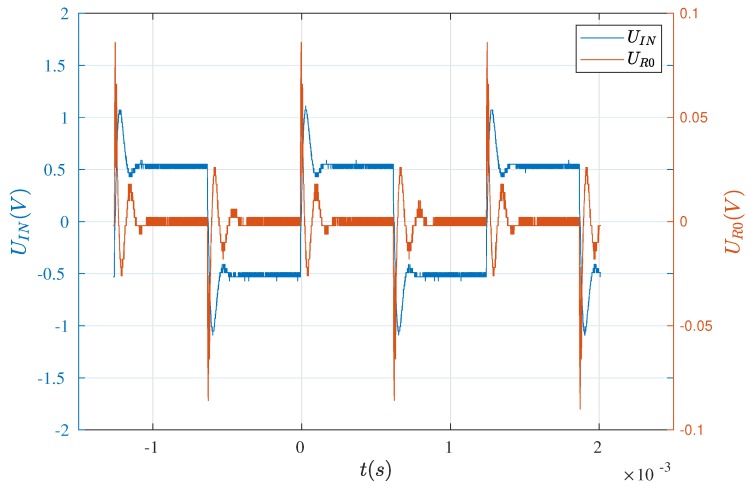
Waveforms recorded by the means of the oscilloscope.

**Figure 5 sensors-19-00255-f005:**
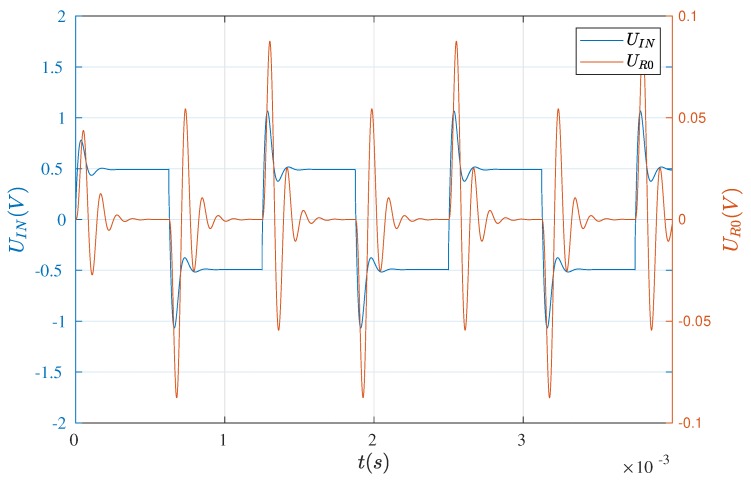
Waveforms obtained on the basis of the model.

**Figure 6 sensors-19-00255-f006:**
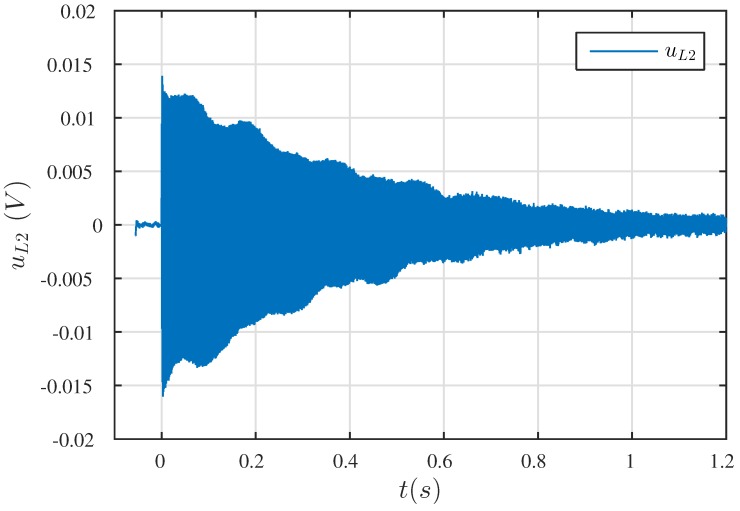
Waveforms of the wire’s natural frequency.

**Figure 7 sensors-19-00255-f007:**
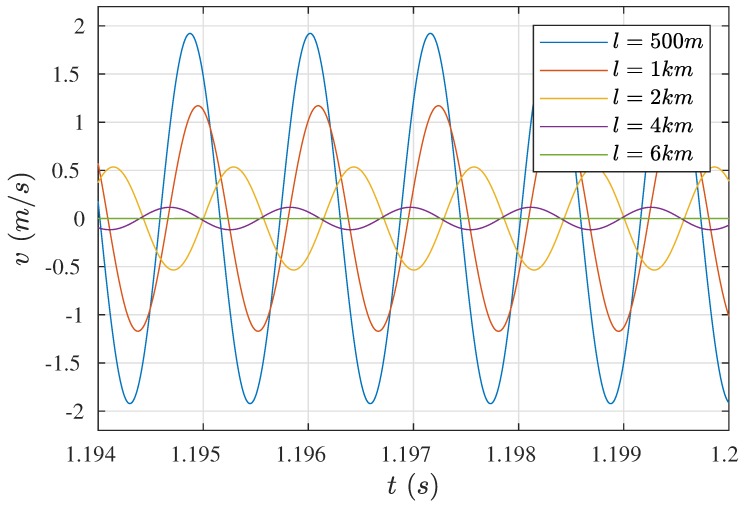
Waveforms of the velocity of the wire middle point (reduced mass) for various lengths of cables.

**Figure 8 sensors-19-00255-f008:**
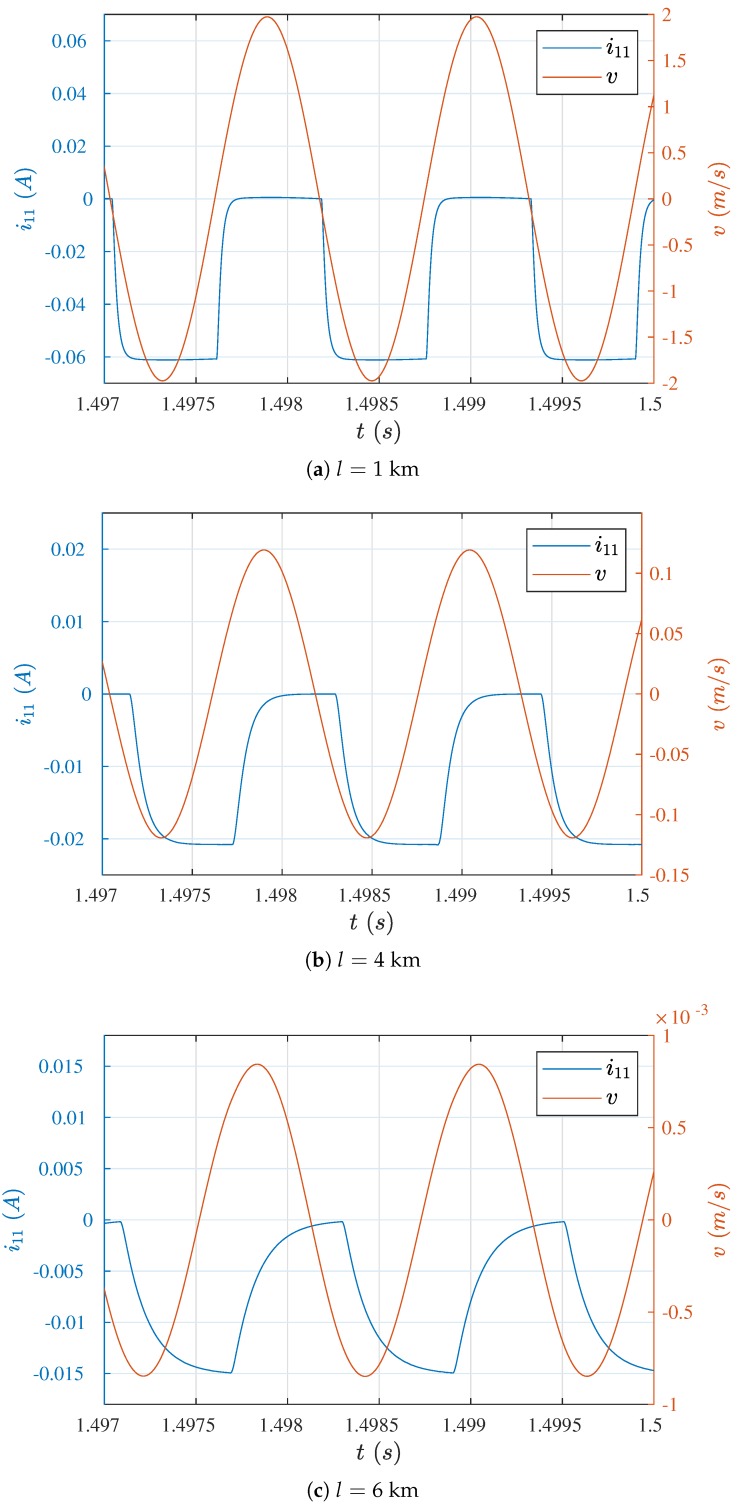
Waveforms of current of the drive coil $L_{1}$ and the velocity of the wire middle point for various lengths of cables.

**Figure 9 sensors-19-00255-f009:**
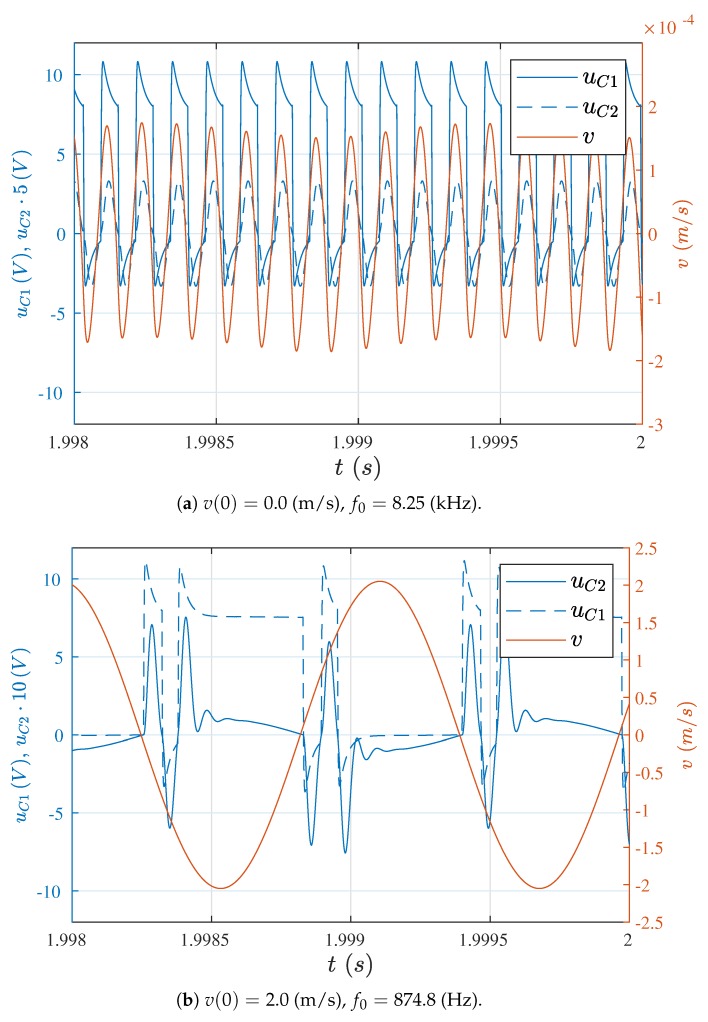
Voltage waveforms on capacitors for l=250 m and $M_{12}=M_{12-real}$.

**Figure 10 sensors-19-00255-f010:**
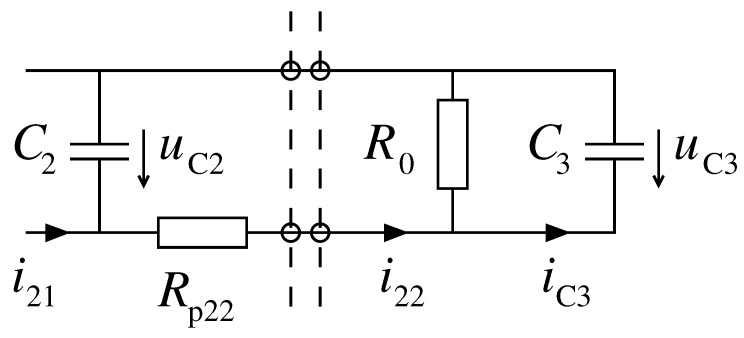
Placement of the filtering capacitor $C_{3}$.

**Figure 11 sensors-19-00255-f011:**
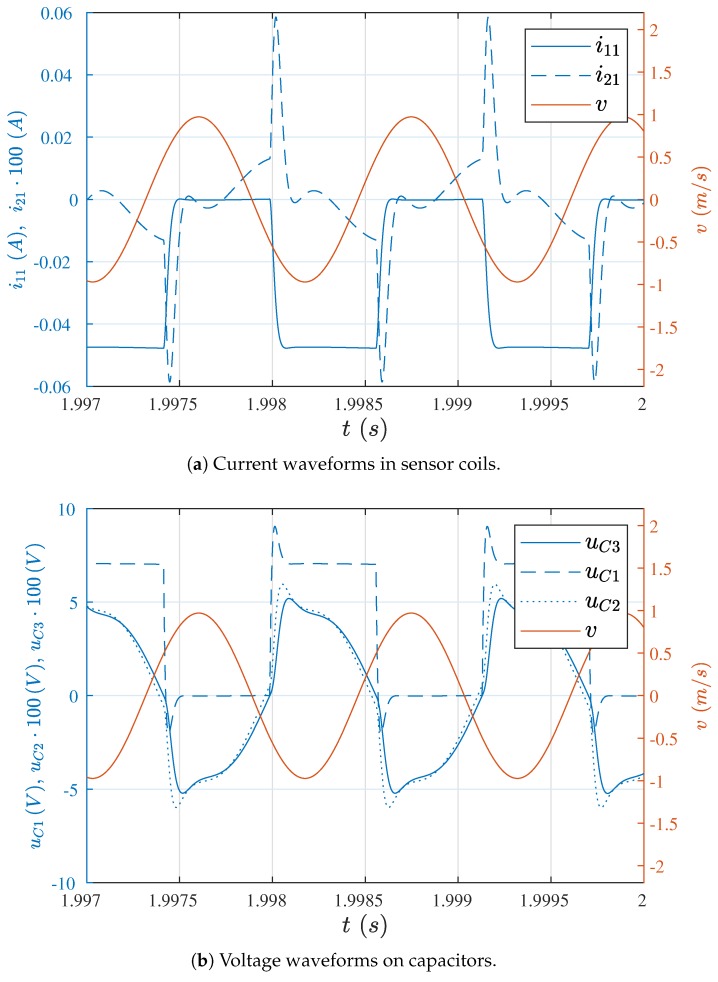
Waveforms for l=1 km, $C_{3}$ = 470 nF, and $M_{12}$ = 0.3$M_{12-real}$.

**Figure 12 sensors-19-00255-f012:**
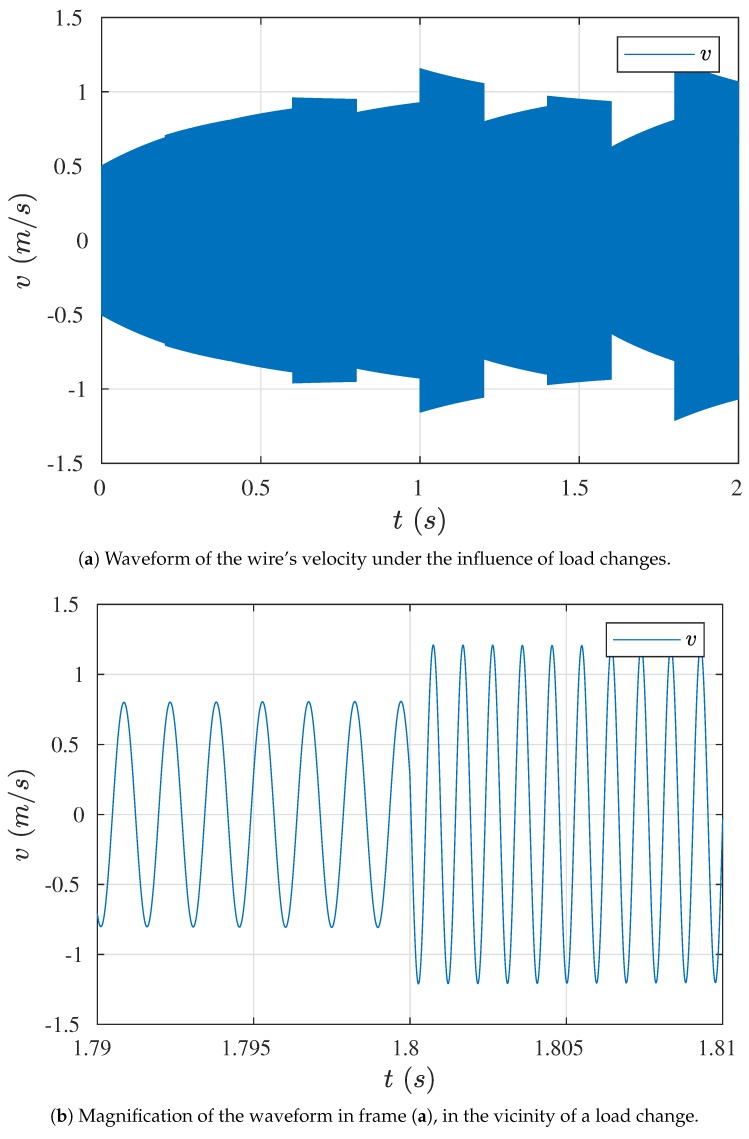
The wire’s response on step load changes.

**Figure 13 sensors-19-00255-f013:**
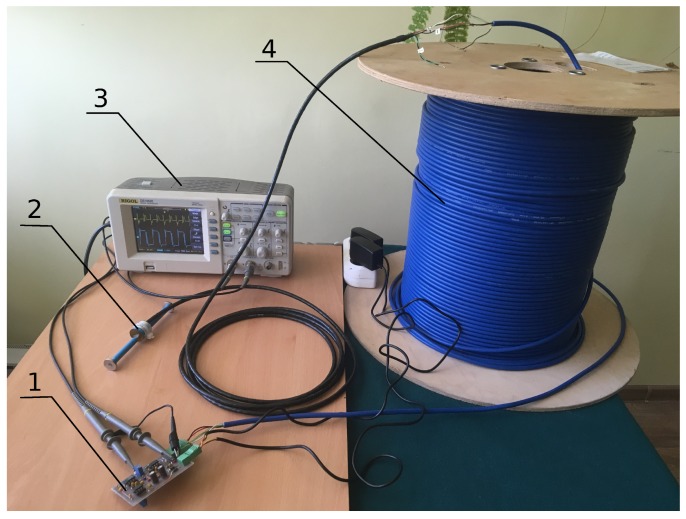
Photograph of the research setup: 1–impulse exciter (electronic components mounted on a Printed Circuit Board), 2–wire sensor, 3–oscilloscope, and 4–electrical cables (500 m).

**Figure 14 sensors-19-00255-f014:**
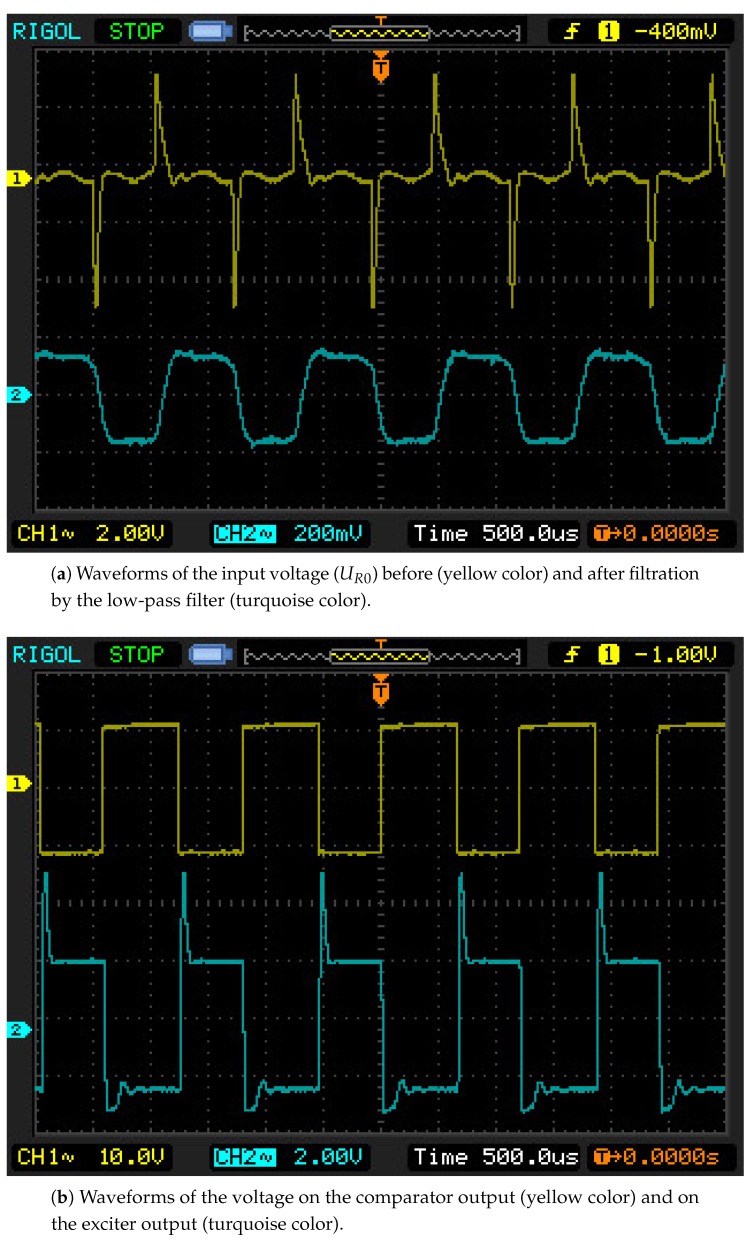
Oscillographs for cables of a length l=1.5 m.

**Figure 15 sensors-19-00255-f015:**
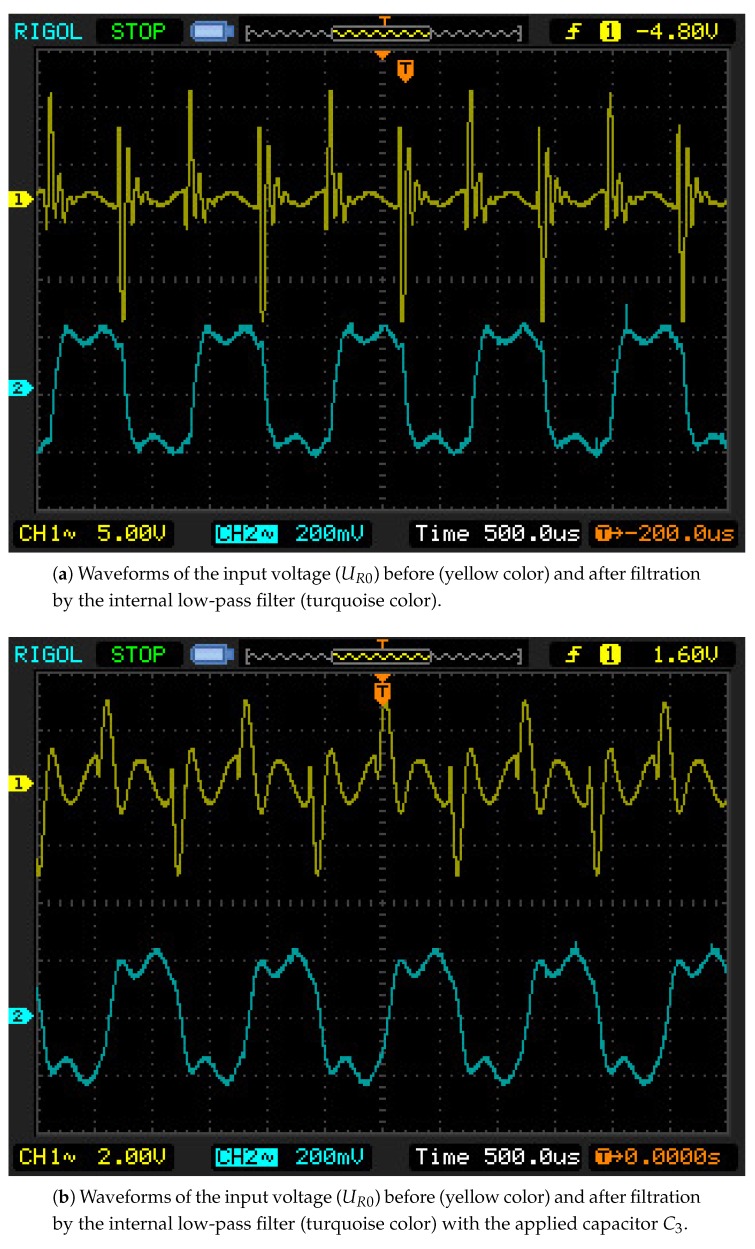
Oscillographs for cables of a length l=160 m.

**Figure 16 sensors-19-00255-f016:**
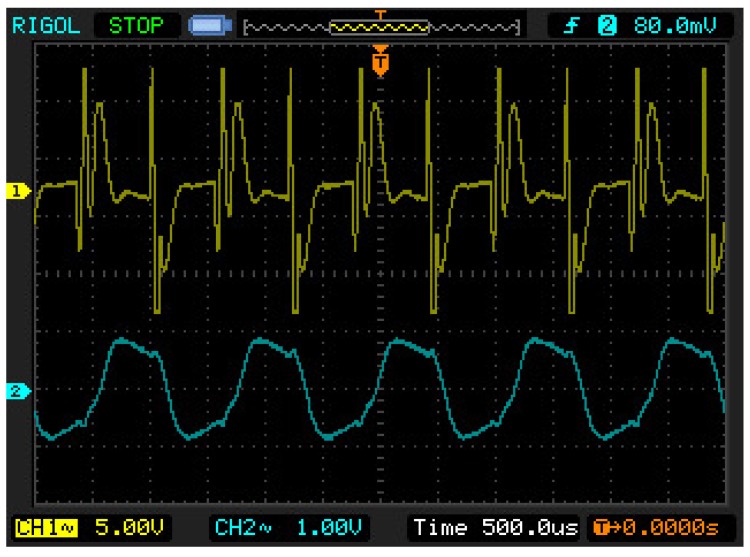
Oscillographs for cables of a length l=500 m. Waveforms of the input voltage ($U_{R0}$) before (yellow color) and after filtration by the internal low-pass filter (turquoise color) with the applied capacitor $C_{3}$.

**Table 1 sensors-19-00255-t001:** Parameters of the equivalent model for the 500 m long electric cable.

Parameter	Value	Unit
$L_{1}$	4.63	mH
$L_{2}$	4.63	mH
$M_{12}$	0.37	mH
$C_{1}$	104.9	nF
$C_{2}$	58.8	nF
$R_{1}$	93.4	Ω
$R_{2}$	93.1	Ω
$R_{p11}$	27.1	Ω
$R_{p12}$	27.1	Ω
$R_{p21}$	27.1	Ω
$R_{p22}$	27.1	Ω
$R_{d}$	303.5	Ω
$R_{0}$	9.85	kΩ
*E*	2.0	V
*f*	800	Hz
$\frac{\partial L_{1}\left(x\right)}{\partial x},\frac{\partial L_{2}\left(x\right)}{\partial x}$	−0.166	H/m

One of the terminals of condenser $C_{1}$ was connected to the foil shielding of the bunched wires.

**Table 2 sensors-19-00255-t002:** Comparison of the theoretical values ($v_{0-eq}$) with the values obtained on the basis of computer simulations ($v_{0-sym}$).

l(m)	$i_{110}$ (mA)	$v_{0-\mathrm{sym}}$ (m/s)	$v_{0-\mathrm{eq}}$ (m/s)	Difference in %
500	61.1	1.97	2.07	5.1
1000	47.8	1.20	1.27	5.8
2000	33.4	0.55	0.62	12.7
3000	25.6	0.27	0.36	33.3
4000	20.8	0.12	0.24	100
6000	–	–	0.0009	–

**Table 3 sensors-19-00255-t003:** Cycle of changes in the wire’s equivalent coefficient of elasticity.

*k* (N/m)	558.31	586.23	502.48	642.06	446.65
$f_{0}$ (Hz)	874.63	896.23	829.75	937.94	782.29
*k* (N/m)	697.89	390.82	753.72	334.98	809.55
$f_{0}$ (Hz)	977.87	731.77	1016.2	677.49	1053.2

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
