# Peer review of "Measurements of Dynamic Deformations of Building Structures by Applying Wire Sensors"

_sensors, 2019, doi:10.3390/s19020255_

Reviewer 1 Report

The work presented in this paper regarding vibrating wire sensor focusing on the structure of sensor and the length of electric cables is very useful in practical measurement of related areas. Tests and experiments are well presented. However, my concerns are as follows:

1.     The work is lack of technical depth. Since electromagnet circuity are very mature techniques, just to find the suitable filtering components to make a good matching are not quite novel contributions.

2.     The description of related work is not comprehensive. Only a few related work/techniques are described. Investigations of the very close area about vibrating wire sensor performance are not quite well described.

3.     Please indicate in Figure 14 which is vibration sensor, and which is electric cable.

4.     Line 105, 133, line 231 symbol “÷” should be “-”?

Author Response

The work is lack of technical depth. Since electromagnet circuity are very mature techniques, just to find the suitable filtering components to make a good matching are not quite novel contributions.

Of course, the theory of discrete electrical systems, including the theory of two-port network and the coupling between electrical and mechanical elements is well known, so the work does not bring much to the point.

However, its main goal was to answer the question: what maximum length of cables can we use to ensure correct operation of the wire sensor using a new type of impulse exciter?

The impulse exciter that was described in the work is a device with a different manner of operation - in relation to the solutions used so far – by what transferring the existing knowledge about the length of cables would be a very risky choice.

The work gives an answer to this question and this should be considered as a contribution to the development of the issue.

The description of related work is not comprehensive. Only a few related work/techniques are described. Investigations of the very close area about vibrating wire sensor performance are not quite well described.

The work is devoted to the problem of dynamic measurements with the use of wire sensors, therefore the influence of the length of connecting wires and mutual inductance on their proper operation have been emphasized. For this reason, the content regarding the construction and applications of the devices have been more modestly treated.

Also, in previous works, author touched on the subject of construction and applications, so he did not want to repeat the given content again. Author introduced however a short text regarding the need for dynamic measurements and two new references to related works.

Please indicate in Figure 14 which is vibration sensor, and which is electric cable.

The drawing has been improved.

Line 105, 133, line 231 symbol “÷” should be “-”?

The text has been corrected.

Reviewer 2 Report

The text of the manuscript focuses on the interesting part and field of applied research. It is treated with certain formal shortcomings. For  example, poorly processed images Figure 2 - image quality, Figure 4,  Figure 11 - font size, missing symbol description in Figure 4, paramter  symbols both italic and perpendicular (Figure 4, Table 1 ... etc) Figure 5-10, 12 are less clearly described by reference to the article text. The  axes x and their values in Figures 8, 9, 10, 12 are unnecessarily  burdened by displacement offset, there are changes in the area ms. For the reader, this entry may be confusing.

From  the content page, it is unclear to build a relationship model (1) - (5)  because the figure 4 model is a model of electrical conduction as only  RC structure, which is a deep error in the field of theoretical  electrotechnics for the described case of excitation by a non-harmonic  signal. This section recommends correcting and changing the model, which will then result in changes in the text of the conclusion.

It might be worthwhile to acknowledge the support of the research, to include pictures 15-17 outside the reference text.

After overworking, the text of the work may cause an interesting message to the professional community.

Author Response

The text of the manuscript focuses on the interesting part and field of applied research. It is treated with certain formal shortcomings. For example,

poorly processed images Figure 2 - image quality,

=> The drawing has been improved.

Figure 4, Figure 11 - font size,

parameter symbols both italic and perpendicular (Figure 4, Table 1 ... etc.)

=> Figures have been improved.

missing symbol description in Figure 4,

=>Description of the symbols has been introduced

The axes x and their values in Figures 8, 9, 10, 12 are unnecessarily burdened by displacement offset, there are changes in the area ms. For the reader, this entry may be confusing.
=>This displacement offset is important (although the reviewer finds it confusing), because it shows the time after which the system gets a steady state. All simulations at work keep the same timeline so you can compare phase shifts. With regard to this remark, I would like to leave the figures unchanged.

Figure 5-10, 12 are less clearly described by reference to the article text.

=> I do not have much to say about this topic. I have to rely on the reader's imagination.
From the content page, it is unclear to build a relationship model (1) - (5) because the figure 4 model is a model of electrical conduction as only RC structure, which is a deep error in the field of theoretical electrotechnics for the described case of excitation by a non-harmonic signal. This section recommends correcting and changing the model, which will then result in changes in the text of the conclusion.

=> Of course, a model that would take into account the longitudinal inductance of the two-port network (describing the electric cable) would be more relevant to reality. However, this inductance was deliberately omitted (as insignificant) on the basis of the measurements carried out. Impedance measurements indicated clearly the capacitive character of the cable.

Confirmation of the correctness of the adopted model can be the results presented in Figures 5 and 6, showing the real waveforms and obtained on the base of the model.

In my opinion, the consistency of the results is sufficient enough that the RC two-port network model can be considered as reliable for this particular system.

One should also not be afraid of nonharmonic excitations, because the presented results relate to excitation of a square wave form with very high dynamics.

It might be worthwhile to acknowledge the support of the research, to include pictures 15-17 outside the reference text.

=>The pictures 15-17 have been moved to the right places.

Reviewer 3 Report

Comments to the authors:

I suggest same minor corrections.

Page 2: Introduction: Line 55: Authors should include: “Another solution for vibrating measurement is high precision impedance-frequency transducers (capacitive or inductive) which using quartz crystals which compensate temperature drift, and have high stability and fast response, and can be modulated with very low frequency. These methods are shown in ref.:

-Matko V., Jezernik K. Greatly improved small inductance measurement using quartz crystal parasitic capacitance compensation. Sensors, 2010, 4, 10, 3954-3960, doi: 10.3390/s100403954

-Matko V. Next generation AT-cut quartz crystal sensing devices. Sensors 2011, 5, 4474-4482.

doi: 10.3390/s110504474.

Authors should include the above mentioned references in the manuscript.

Page 3: Fig. 2. Should be better quality. The switching part should be more detailed.

Pages 6, 9, 10, 11, and 12: On Figures: the legends are not clear which is which signal?

Page 16: Fig. 15, 16, and 17 are inserted in chapter References ????

Page 15: Fig. 14 needs more description and should be more clear, and needs explanation what is presented on the Figure.

Page 13: Conclusion: What are the main odvantages of the proposed method?

Could authors write something about dynamic deformation of building structures, because of the title of the manuscript.

Author Response

@page { margin: 2cm } p { margin-bottom: 0.25cm; line-height: 115% }

Page 2: Introduction: Line 55: Authors should include: “Another solution for vibrating measurement is high precision impedance-frequency transducers (capacitive or inductive) which using quartz crystals which compensate temperature drift, and have high stability and fast response, and can be modulated with very low frequency. These methods are shown in ref.:

-Matko V., Jezernik K. Greatly improved small inductance measurement using quartz crystal parasitic capacitance compensation. Sensors, 2010, 4, 10, 3954-3960, doi: 10.3390/s100403954

-Matko V. Next generation AT-cut quartz crystal sensing devices. Sensors 2011, 5, 4474-4482.

doi: 10.3390/s110504474.

Authors should include the above mentioned references in the manuscript.

=> References have been entered into the text.

Page 3: Fig. 2. Should be better quality. The switching part should be more detailed.

=> The Fig.2  has been improved. With reference to the switching system, I added a reference to the proper publication and short information about the switching transistor.

Pages 6, 9, 10, 11, and 12: On Figures: the legends are not clear which is which signal?

=>In my opinion, the red and blue colours clearly differentiate the charts. Maybe the reviewer received a black and white version of the work?

Page 16: Fig. 15, 16, and 17 are inserted in chapter References ????

=>The figures have been moved to the right places.

Page 15: Fig. 14 needs more description and should be more clear, and needs explanation what is presented on the Figure.

=>Fig.14 has been described in more detail.

Page 13: Conclusion: What are the main odvantages of the proposed method?

The main advantage of the work is to indicate the most important factors affecting the difficulties in conducting dynamic measurements using wire sensors. The example of a impulse exciter has demonstrated the existence of a critical length of connecting cables above which the device will not take action. The high influence of sensor coils coupling on the correctness of the impulse driver operation was also demonstrated. It should be noted that the impulse exciter presented in the paper has very good resistance to changes in load which, compared to the solutions available on the market, makes it a more practical solution.

Could authors write something about dynamic deformation of building structures, because of the title of the manuscript.

I introduced a short text regarding the need for dynamic measurements - lines 36-41.